# UBE2G1 Is a Critical Component of Immune Response to the Infection of *Pseudomonas Plecoglossicida* in Large Yellow Croaker (*Larimichthys crocea*)

**DOI:** 10.3390/ijms23158298

**Published:** 2022-07-27

**Authors:** Jia Peng, Wanbo Li, Bi Wang, Sen Zhang, Yao Xiao, Fang Han, Zhiyong Wang

**Affiliations:** 1Key Laboratory of Healthy Mariculture for the East China Sea, Ministry of Agriculture and Rural Affairs, Fujian Provincial Key Laboratory of Marine Fishery Resources and Eco-Environment, Fisheries College, Jimei University, Xiamen 361021, China; 202011710010@jmu.edu.cn (J.P.); li.wanbo@jmu.edu.cn (W.L.); 201361000026@jmu.edu.cn (B.W.); 202012951009@jmu.edu.cn (S.Z.); 202012951035@jmu.edu.cn (Y.X.); 2Laboratory for Marine Fisheries Science and Food Production Processes, Qingdao National Laboratory for Marine Science and Technology, Qingdao 266237, China

**Keywords:** *Larimichthys crocea*, internal organ white-spot disease, ubiquitination, ubiquitin-binding enzyme E2G1 (UBE2G1)

## Abstract

The large yellow croaker (*Larimichthys crocea*) is one of the most economically valuable mariculture fish in China. Infection of *Pseudomonas plecoglossicida* can cause an outbreak of “internal organ white-spot disease”, which seriously affects the aquaculture of the large yellow croaker. Ubiquitylation is closely related to the post-translation modification of proteins and plays a vital role in many hosts’ immune defense pathways, while the E2-binding enzyme is a key factor in ubiquitination. Our previous genome-wide association study found that the ubiquitin-binding enzyme E2G1 (designed *LcUbe2g1*) was one of the candidate genes related to disease resistance in large yellow croaker. In this study, we analyzed the molecular characteristics, function, and immune mechanism of the *LcUbe2g1*. The full-length cDNA is 812 bp, with an open reading frame of 513 bp, encoding 170 amino acid residues. The results of the RT-qPCR and immunohistochemistry analysis revealed that its transcription and translation were significantly activated by the infection of *P. plecoglossicida* in large yellow croaker. Immunocytochemistry experiments verified the co-localization of *Lc*UBE2G1 and the ubiquitin proteins in the head kidney cells of large yellow croaker. Through GST pull-down, we found that *Lc*UBE2G1 interacted with NEDD8 to co-regulate the ubiquitination process. The above results indicate that *Lc*UBE2G1 is essential in the regulation of ubiquitination against *P. plecoglossicida* infection in large yellow croaker, which lays a foundation for further study on the resistance mechanism of internal organ white-spot disease.

## 1. Introduction

Large yellow croaker (*Larimichthys crocea*) is one of the most economically valuable mariculture fish in the eastern and southern coastal waters of China [1,2,3]; its healthy farming is very important to aquaculture farmers. However, bacterial diseases have hindered the rapid development of the large yellow croaker aquaculture industry [4]. The “internal organ white-spot disease” is a highly prevalent and immensely destructive disease in the cultured large yellow croaker, and *Pseudomonas plecoglossicida* exists as its main pathogenic bacteria [5].

Ubiquitin (Ub) has been considered as one of the most important regulatory proteins in eukaryotes since it was first isolated by Gideon Goldstein and colleagues in 1975 [6]. In three billion years of evolution, this remarkable eukaryotic protein has barely changed. This 8 kDa polypeptide is essential for the ubiquitination of cells [7], which normally break down the damaged or unwanted proteins in order to recycle the building blocks of amino acids [8]. The degradation proteins are first tagged with a chain of ubiquitin protein, which is a multi-step catalytic reaction mediated by a ubiquitin-activating enzyme (E1), a ubiquitin-conjugation enzyme (E2), and a ubiquitin-protein ligase (E3) [8]. Ubiquitin-mediated proteolysis plays a critical role in a large quantity of cellular functions, such as cell cycle regulation, protein quality control, cytokine signal transduction, stress responses, and cellular homeostasis [9,10]; defects in this system also lead to cancer and neurodegenerative diseases [10].

The E2-conjugating enzyme, as a pivotal factor in ubiquitination, serves as the key regulator in the topology of the Ub chain and the choice of E3 [11]. In recent years, growing evidence has shown that they were also associated with the maladjustment of the tumor microenvironment and were involved in various tumor-promoting processes, including DNA repair, apoptosis, cell cycle progression, tumorigenesis, and signaling pathways [12]. For instance, Somasagara et al., found that UBE2B could promote the stemness of ovarian cells through stabilizing *β*-catenin [13]. Feng et al., suggested that UBE2J1 negatively regulated type I IFN expression, facilitating RNA virus replication by mediating the ubiquitination and degradation of transcription factor IRF3 [14]. Fujita et al., found that UBE2C could change the mitotic population and influence their expansion and invasion [15]. Liu et al., have illustrated that UBE2G1 was involved in the innate immune response of oysters to invasive pathogens [16]. However, the underlying mechanism of *Lc*UBE2G1 against pathogen invasion remains unclear. In this study, we analyzed the molecular role, function and immune mechanism of *Lc*UBE2G1, and suggest that *Lc*UBE2G1 is crucial for the innate immune defense against *Pseudomonas plecoglossicida* in the large yellow croaker.

## 2. Results

### 2.1. Sequence Characteristics of LcUbe2g1

The cDNA sequence of *LcUbe2g1* was obtained from our transcriptomic database. The full-length cDNA of *LcUbe2g1* is 812 bp and contains an open reading frame of 513 bp encoding a polypeptide of 170 amino acid residues (Figure 1A), a 266 bp 5′ untranslated region (UTR) and a 33 bp 3′ UTR. The tertiary structure prediction results showed that *Lc*UBE2G1 contains a UBC core catalytic domain consisting of four α helices and four β sheets, a conserved HPN motif (His81, Pro82, Asn83) and a cysteine (Cys90) active site, suggesting *Lc*UBE2G1 was similar to most of E2s (Figure 1B).

Multiple alignment of the amino acid sequences indicated that the sequence of *LcUbe2g1* was highly conserved (Figure 2A). The predicted molecular mass of the *Lc*UBE2G1 polypeptide was 19.45 kDa, and the isoelectric point was 5.34. The phylogenic analysis revealed that *LcUbe2g1* is widely distributed in most living organisms, including shellfishes, fishes, birds, amphibians and mammals. The protein sequence of *Lc*UBE2G1 was clustered with other common bony fishes and had the closest phylogenetic relationship with *Collichthys lucidus*, and the phylogenetic relationships were consistent with their taxonomic classification (Figure 2B).

**Figure 2 ijms-23-08298-f002:**
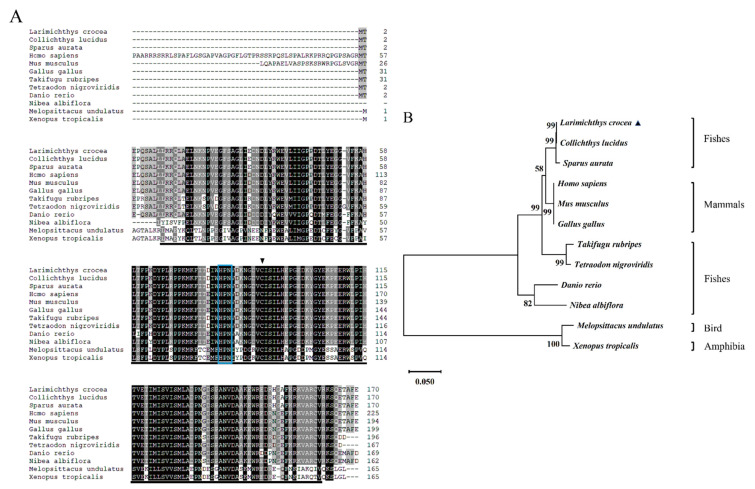
Sequence alignment and phylogenetic analysis of *LcUbe2g1* and its homologs. (**A**) Multiple sequence alignment of *LcUbe2g1* and its homologs. The blue boxes indicate the ubiquitin (Ub) binding motifs (HPN), the black triangle indicates the active site cysteine (C), the black shading and gray shading indicate the amino acid sequence similarity of 100% and 80%, respectively, and the GenBank accession numbers of amino acid sequences of the species used are detailed in Table 1; (**B**) Phylogenetic analysis of *LcUbe2g1* and its homologs. The number at each node indicates the percentage of bootstrapping after 1000 replications. Scale bar refers to evolutionary distance of 0.05 million years.

### 2.2. Tissue Expression and Subcellular Localization Analysis

To validate the tissue-specific pattern of *LcUbe2g1*, the expression of *LcUbe2g1* in the tissues of large yellow croaker was detected by RT-qPCR. As shown in Figure 3A, *LcUbe2g1* was ubiquitous in all of the tissues tested (kidney, gills, fins, head kidney, brain, spleen, liver and intestine), with the highest level in the brain, followed by kidney, gills, head kidney, spleen, and liver.

To further investigate the subcellular localization of *Lc*UBE2G1, two expression vectors, pEGFP-N1 and pEGFP-N1-*LcUbe2g1*, were transiently transfected into the head kidney cells of large yellow croaker, respectively. The fluorescent microscope imaging revealed that the localization of pEGFP-N1-*LcUbe2g1* protein was same as the pEGFP-N1 (control), expressed in both the nucleus and cytoplasm (Figure 3B).

### 2.3. Expression Analysis of LcUbe2g1 mRNA and Protein under P. plecoglossicida Infection

To study the impact of the *P. plecoglossicida* infection on the expression of *LcUbe2g1*, the temporal expression in the liver, kidney, and spleen was detected by RT-qPCR at 3, 6, 12, 24, 48, 72, 96 and 120 h after infection. As shown in Figure 4A, the expression of *LcUbe2g1* was significantly different between the experimental and control groups after infection with *P. plecoglossicida.* In the kidney, a significant upregulation of *LcUbe2g1* was observed at 96 h after infection. In the liver, the relative expression level of *LcUbe2g1* showed a gradual upward trend with the prolongation of infection time, and reached the maximum at 72 h after infection, and the expression level began to gradually decrease afterwards.

To further validate the results of RT-qPCR, the spleen, kidney, liver and brain tissues of large yellow croaker were sampled at 96 h after *P. plecoglossicida* infection, while the tissues of healthy fish were sampled as control; immunohistochemical staining was then carried out to examine the changes in the distribution of the UBE2G1 protein in these tissues. The results showed that the expression of the *Lc*UBE2G1 protein in the four, tested tissues was significantly enhanced after *P. plecoglossicida* stimulation (Figure 4B). Thus, UBE2G1 played an active role at both the transcriptional and translational levels during the immune response to *P. plecoglossicida* stimulation in large yellow croaker.

### 2.4. Recombinant LcUBE2G1 Protein

After IPTG induction, the whole-cell lysate of *E. coli* BL21 (DE3), containing the recombinant protein pGEX-6P-1-*Lc*UBE2G1, was separated by SDS-PAGE, and a distinct band with a molecular weight of ∼45 kDa was observed (Figure 5). As expected, a single band of 19 kDa and a single band of 45 kDa can be observed after cleavage of the GST tag with PreScission Protease, representing the purified *Lc*UBE2G1 protein, GST-tagged protein and GST-*Lc*UBE2G1 recombinant protein (Figure 5).

### 2.5. Interaction between LcUBE2G1 and Ub

The localization of the *Lc*UBE2G1 protein and Ub protein in the head kidney cells was identified through confocal scanning microscope. The result unraveled the co-localization of the Ub protein (red fluorescence) and *Lc*UBE2G1protein (green fluorescence) on the cell membrane, indicating that *Lc*UBE2G1 might interact with Ub (Figure 6).

### 2.6. Characterization of the LcUBE2G1 Interacting Proteins

To further study the interaction between Ub and *Lc*UBE2G1, GST pull-down, SDS-PAGE separation (Figure 7A) and liquid chromatography–mass spectrometry (LC-MS) experiments (Figure 7B) were performed. We did not find any Ub in the eluate, suggesting that the two proteins did not interact in these conditions. Additionally, we identified 184 potential *Lc*UBE2G1 interacting proteins. As shown in Figure 7C, in the highly connected network composed of 53 proteins from the mass spectrometry results (disconnected nodes are not shown), the NEDD8 protein appeared to be a key protein which directly interacted with *Lc*UBE2G1.

## 3. Discussion

As with the immune system in human, the fish immune system has a constant battle with invading pathogens. Ubiquitination is not only involved in the recognition and elimination of pathogens, microbial foreign bodies and the innate immunity required by damaged cells in different cells, receptors and signaling pathways of the immune system, but also participates in the post-translational modification of proteins in immune-signaling pathways. This ubiquitin modification is a key reason for the complexity, diversity and specificity of the immune response [17,18]. Ubiquitin is an energy-dependent post-translational modification process in which the 8 kDa ubiquitin protein covalently attaches to one or more lysine residues of the substrate protein. This process consists of three successive events: activation (E1); conjugation (E2); and ligation (E3) of Ub protein.

There are two E1 (UBA1 and UBA6), 38 E2 and more than 600 E3 enzymes in the human genome [19]. The E1 enzyme activates ubiquitin via an ATP-dependent reaction to form a thioester bond, which is then transferred to cysteine, the active site of the ubiquitin-binding enzyme E2, the latter in turn forms an isopeptide bond and cooperates with E3s to modify the target protein [20]. In mammals, growing research confirms that the ubiquitin system is closely associated with the regulation of the host immune defense pathways [21]. For example, the ubiquitin system mediates the negative regulation of the inflammasomes, which is involved in the binding of autophagosomes to bacterial targets, effectively preventing unnecessary inflammasomes from causing excessive inflammation and cell death [22]. Recently, Liu et al., demonstrated the importance of Ube2g1 in the innate immune system of oysters [16]. Although studies have confirmed that NRDP1 (an E3 ubiquitin ligase) is involved in the immune system of large yellow croaker [3], the role of the ubiquitin system in the immune response against tongue-scale infection remains unclear in large yellow croaker.

In previous research of our laboratory, a *P. plecoglossicida* challenge experiment was conducted on large yellow croaker. A total of 769 differentially expressed genes were detected by transcriptomic analysis of spleen samples with a significant difference in the bacterial load. Among these differentially expressed genes, *LcUbe2g1* was significantly upregulated. The *LcUbe2g1* gene was cloned and characterized for the first time in large yellow croaker. Similar to members of the mammalian E2G1 family, *LcUbe2g1* has a core catalytic domain (UBC), consisting of a highly conserved HPN sequence and a single active site responsible for Ub reception and transmission [23,24]. The phylogenetic analysis showed that the *LcUbe2g1* sequence was consistent with the classification of other true-eared bats, and its phylogeny corresponded to the traditional taxonomy.

In addition, the co-localization of *Lc*UBE2G1 and Ub was observed in the cell membrane of the head kidney cell of large yellow croaker, which is consistent with Liu’s research in the Pacific oysters [16]. These findings suggested that *Lc*UBE2G1 might interact with Ub. The upregulation of *LcUbe2g1* at transcriptional and translational level in response to *P. plecoglossicida* infection suggested its active role in the response to pathogen. *LcUbe2g1* was widely expressed in all of the tissues tested; the highest expression was in the brain, followed by the kidney, and the lowest was in the intestine. This is similar to the tissue expression profile of UBC9 (a member of the E2 family) in large yellow croaker and semi-smooth tongue [25]. The study also showed that neurodegeneration is the most common disorder of all of the disease types caused by dysfunction of the ubiquitin system [26]. These results suggested that *Lc*UBE2G1 played different defense roles in different biological processes. A recent study found a key mutation in UBE2G1 in the brains of AD patients [27]. UBE2G1 also plays an important role in regulating the destruction of carbon microstructure substrates [8]. This confirms that the high expression of *LcUbe2g1* in the brain is associated with brain-related diseases. In addition, the kidney is an important immune organ of the large yellow croaker. The high expression of *LcUbe2g1* in the kidney confirmed its important role in the immune defense in large yellow croaker.

In vitro ubiquitination experiments on *Ube2g1* from Pacific oysters have confirmed that the binding between UBE2G1 and Ub does not depend on E1 [16]. To further understand the specific mechanism of *LcUbe2g1*, a GST-tagged *Lc*UBE2G1 recombinant protein was constructed, the interacting proteins of *Lc*UBE2G1 were validated by the GST pull-down. In addition to Ub, other proteins that interact with *Lc*UBE2G1 were found. The most similar NEDD8 to ubiquitin was found by proteomics and STRING analysis. NEDD8 is highly conserved in most eukaryotes and has a regular function as a covalent modifier, acting on Cullin family members through an enzyme-linked reaction similar to ubiquitination; cellular protein degradation is regulated by this reaction [9,28].

The Cullin family proteins are the scaffold components of skp1cullinf-box ubiquitin ligase complex. Under the regulation of NEDD8, they control the ubiquitination and proteasome degradation of cell-cycle regulatory protein p27 and cell-cycle protein E. Studies on the components of the NEDD8 pathway in Schizosaccharomyces, Caenorhabditis elegans, Drosophila and mice indicated that NEDD8 played an important role in developmental regulation, cell viability and growth [29]. Gao et al.’s study also confirmed that NEDD8 can negatively regulate NF-kB [30]. Thus, the ubiquitin and NEDD8 pathways always seem to work closely together to control protein function.

## 4. Materials and Methods

### 4.1. Pathogen Stimulation and Tissue Collection

The healthy large yellow croaker (weight 28.5 ± 8.5 g) purchased from Ningde, Fujian, China were first fed with commercial feed for 10 days in a 4 m^3^ tanks (25–26 PSU, 23–26 °C), and the head kidney, spleen, muscle, kidney, intestine, liver, gills and brain tissues of three healthy large yellow croaker were collected. Subsequently, *P. plecoglossicida* (1.0 × 106 cfu/mL) were added to the tank and the liver, spleen and kidney were gathered sequentially at 3, 6, 12, 24, 48, 72, 96 and 120 h after stimulation, all of the tissues were rapidly frozen in liquid nitrogen and then stored at −80 °C for RNA extraction.

### 4.2. RNA Extraction and cDNA Synthesis

The total RNA was extracted from 100 mg collected tissues, according to the instructions of the TransZol Up Plus RNA Kit (TransGen Biotech, Shanghai, China). The extracted RNA samples were treated with DNase I (Takara, Beijing, China) for 20–30 min to remove genomic DNA. Its quality and concentration were detected by agarose gel electrophoresis and microplate reader, Multiskan Go (Thermo, Waltham, MA, USA). The first-strand cDNA was obtained by reverse transcription, using the GoScript^TM^ Reverse Transcription System (Promega, Madison, WI, USA), and the effect of the reverse transcription was tested by PCR with internal reference gene, *β-actin*. The extracted cDNA was stored at −20 °C for subsequent use.

### 4.3. Cloning and Sequence Analysis of LcUbe2g1

The full-length cDNA sequence of *LcUbe2g1* (GenBank ID: ON081958.1) was intercepted from the transcriptomic database in our laboratory. A pair of specific primers, P1 and P2 (Table 2), were designed and the complete cDNA fragment of *LcUbe2g1* was cloned using the high-fidelity enzyme 2 × Phanta Max Master Mix (Vazyme, Nanjin, China). The reaction conditions for the polymerase chain reaction (PCR) are as follows: 95 °C for 3 min; 35 cycles of 95 °C for 15 s; 55 °C for 20 s; 72 °C for 45 s.

Sequence homology analysis was carried out using the BLAST program (http://blast.ncbi.nlm.nih.gov/Blast.cgi). Multiple alignments of *Lc*UBE2G1 amino acid sequences of the large yellow croaker with other species were completed using Clustalx 2.1 and the results were plotted by GeneDoc. A phylogenetic tree of the *Lc*UBE2G1 protein was constructed through MEGA 10.2.6 with the neighbor-joining algorithm. Tertiary structure modeling was performed using Phyre2 (PHYRE2 Protein Fold Recognition Server (ic.ac.UK)) and mapped by VMD 1.9.4.

### 4.4. Construction of Expression Vector

The cDNA fragment encoding the mature peptide of *Lc*UBE2G1 was amplified using two pairs of primers P3, P4 and P5, P6 (Table 2), with specific restriction enzyme sites. We subsequently cloned the fragments into different expression vectors, pGEX-6P-1 and pEGFP-N1 (preserved by our lab), respectively. The positive transformants were screened by PCR with the primers and confirmed by sequencing (Sangon Biotech, Shanghai, China). The pGEX-6P-1-*Lc*UBE2G1 and pEGFP-N1-*Lc*UBE2G1 expression vectors were successfully constructed.

### 4.5. Prokaryotic Expression and Purification of LcUBE2G1

The recombinant plasmid pGEX-6P-1-*Lc*UBE2G1 and the negative control plasmid pGEX-6P-1 were simultaneously transferred into competent cells BL21 (Vazyme, Nanjin, China). The positive recombinant strains were picked from the solid plates and tested by PCR with primer P3 and P4 (Table 2). It was then inoculated in liquid LB medium (50 µg/mL ampicillin, 150 mL) and incubated at 37 °C and 200 rpm until the OD600 of the medium reached 0.6–0.8, the induction agent IPTG (0.1 mM) was added to induce expression for 4 h. The bacterial cells were then immediately collected by centrifugation at 4 °C 12,000× *g* for 10 min, the supernatant was discarded, washed three times repeatedly with ice-cold PBS, the bacterial pellet was resuspended in 10 mL PBS and sonicated on ice for 25 min (total power 35 W). Subsequently, the recombinant protein supernatant was separated from the precipitate by low temperature centrifugation for 15 min (4 °C, 12,000× *g*) and the protein supernatant obtained by sonication was purified using a GST-Tag purification column (Sigma, Madison, WI, USA), according to the manufacturer’s instructions. Finally, the purified protein was detected by 12.5% SDS-polyacrylamide gel electrophoresis, while the remaining purified protein solution was lyophilized into powder form and stored at −80 °C for backup.

### 4.6. Subcellular Localization of LcUBE2G1

The plasmids pEGFP-N1 and pEGFP-N1-*Lc*UBE2G1 were separately transfected into cultured large yellow croaker head kidney cells (Leibovitzs L-15 basal medium, 10% serum, 1% double antibody, 28 °C) by electroporation (BEX CUY21 EDIT II, BEX), then cells were immediately transferred to 12-well plates with sterile coverslips and incubated at 28 °C for 24 h. They were then washed twice with 1 × PBS buffer (pH 7.4), fixed with 4% paraformaldehyde for 10 min at room temperature, permeabilized for 15 min, stained with DAPI (1 µg/mL) for 10 min, and washed three times with 1 × PBS buffer. Finally, the cells were observed and photographed under a fluorescence microscope (Leica SP8, Heidelberg, Germany).

### 4.7. Quantitative Real-Time PCR Analysis of LcUbe2g1

The expression of *LcUbe2g1* mRNA in healthy tissues (kidney, gills, fins, liver, head kidney, brain, intestine and spleen) and its variations in liver, spleen and kidney after *P. plecoglossicida* challenge was assessed using RT-qPCR in the Quant Studio 6 Flex real-time Detection System (Application Biosystems, Waltham, MA, USA). The cDNA from eight healthy tissues and three important immune organs (after *P. plecoglossicida* stimulation) were used as the templates. The *β-actin*-QF and *β-actin*-QR (Table 2) were used to amplify the internal reference gene *β-actin* (152 bp). The specific primers, *LcUbe2g1*-QF and *LcUbe2g1*-QR (Table 2), were used to amplify the cDNA fragment of *LcUbe2g1* (196 bp). The total reaction system (20 µL) was configured according to the Taq Pro Universal SYBR qPCR Master Mix (Vazyme, Nanjin, China) instructions and the qPCR program was set up as follows: 95 °C for 30 s; cycle 40 times at 95 °C for 10 s; and 60 °C for 30 s. The relative expression level of *LcUbe2g1* was analyzed by the comparative Ct method (2^−^^△△Ct^). The SPSS 22.0 software was used for significant difference analysis, considered significant at *p* < 0.05 and extremely significant at *p* < 0.01.

### 4.8. Immunocytochemistry Analysis

The well-grown large yellow croaker head kidney cells were transferred to 12-well plates containing sterile coverslips and incubated at 28 °C for 24 h. The cells were washed twice with 1 × PBS buffer (pH 7.4), fixed with 4% paraformaldehyde for 10 min at room temperature. Following permeabilization with 0.2% Triton X-100 for 15 min, and then treated with 1 × PBS buffer (pH 7.4), the excess Triton X-100 was washed off followed by blocking with 3% BSA for 1 h at 37 °C. The antibody incubation steps are as follows: firstly, the blocked cells were incubated with ubiquitin protein antibody (Mouse; 1:100; Abcam; Cambridge, MA, USA) and UBE2G1 protein antibody (Rabbit; 1:300; Irvine, Gentex Inc., Zeeland, MI, USA) at 37°C for 1 h; then incubated with Alexa Fluor 488-labeled (green) goat anti-rabbit, Alexa Fluor 555-labeled (red) donkey anti-mouse fluorescent secondary antibody (1:500; Beyotime, Shanghai, China) for an additional 1 h at 37 °C. After washing away the excess fluorescent secondary antibody with PBS, the nuclei were stained with 0.2% DAPI for 15 min, washed three times with PBS, and photographed with a fluorescence microscope (Leica SP8, Heidelberg, Germany).

### 4.9. Immunohistochemistry Analysis

The liver, spleen, kidney and brain tissues from large yellow croaker were sampled at the 96th hour after *P. plecoglossicida* infection, while the tissues from healthy fish were sampled as controls and all of the tissues were fixed in 4% paraformaldehyde and subsequently fixed and sectioned in paraffin. After removing excess paraffin from the sections, the sections were incubated in methanol (containing 3% H_2_O_2_) for 25 min at room temperature and blocked with 3% BSA-PBS for 15 min; after washing off the excess blocking solution with PBS, the antibody incubation procedure was as follows, first incubated with anti-UBE2G1 rabbit antibody (1:200) overnight at 4 °C, washed three times with PBS and then incubated with secondary antibody (1:5000 dilution of HPR-labelled goat anti-rabbit antibody) for 40 min at room temperature, finally stained with DAB and hematoxylin for 15 min, and photographed with a fluorescence microscope (Leica SP8, Heidelberg, Germany).

### 4.10. GST Pull-Down Analysis

The purified GST and GST-*Lc*UBE2G1 proteins were separately mixed with glutathione beads and incubated for 1 h at 4 °C. Then, 500 mg of large yellow croaker brain and kidney tissue mixture was homogenized in 5 mL of GST binding buffer (cell lysis buffer for Western blotting and immunoprecipitation PMSF, 100 mM; Beyotime Institute of Biotechnology) on an automatic, fast grinder (JingXin, Shanghai, China). The lysate was then centrifuged at 12,000× *g* for 15 min (4 °C). The supernatant was retrieved and incubated with GST and GST-*Lc*UBE2G1 conjugated glutathione beads at 4 °C for 1 h. After being washed thoroughly with binding buffer (1 × PBS, Ph 7.4), the bound proteins were eluted with elution buffer (20 mM GSH, 50 mM Tris), then analyzed with 12% SDS-PAGE. The differential fragments were excised from the SDS-PAGE gel for mass spectrometry analysis (Shanghai Bio Profile Technology, Shanghai, China).

## 5. Conclusions

In this study, the ubiquitin ligase E2G1 of large yellow croaker was cloned for the first time, and a series of molecular characterization analyses were performed. A related functional analysis of *LcUbe2g1* was also conducted by tissue expression profiling and immunocytochemical experiments. The changes in the tissue expression of *LcUbe2g1* mRNA after infection with *P. plecoglossicida*, and the results of the GST pull-down protein profiling, were also conducive to further clarifying the mechanisms of *Lc*UBE2G1 action in the innate immune defenses of the large yellow croaker. These results suggested that *Lc*UBE2G1 played an important role in large yellow croaker defenses against *P. plecoglossicida* infection.

## Figures and Tables

**Figure 1 ijms-23-08298-f001:**
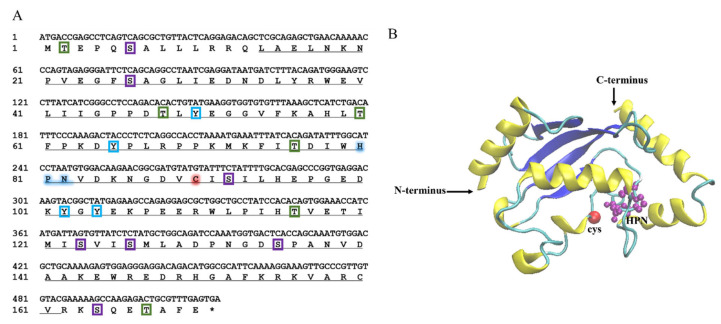
*LcUbe2g1* gene sequence and the tertiary structure of *Lc*UBE2G1 protein. (**A**) The coding sequence of *LcUbe2g1* and deduced amino acid sequence. The stop codons were indicated with asterisk (*), the UBCc (ubiquitin-conjugating catalytic) domain fold was underlined, the blue shade indicate the ubiquitin (Ub) binding motifs (HPN), the red shade indicate the active site cysteine (C), the purple boxes indicate the Ser phosphorylation sites, the green boxes indicate Thr phosphorylation sites and the blue boxes indicate Tyr phosphorylation sites; (**B**) Tertiary structure of *Lc*UBE2G1 protein. *Lc*UBE2G1 contains a UBC core catalytic domain consisting of 4α helices and 4β sheets, the purple spheres represent a conserved HPN pattern (His81, Pro82, Asn83) and the red spheres indicate a cysteine (Cys91) active site.

**Figure 3 ijms-23-08298-f003:**
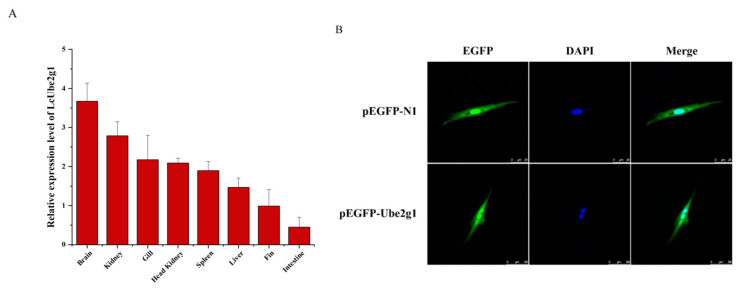
*LcUbe2g1* tissue expression profile and the subcellular localization of *Lc*UBE2G1. (**A**) Relative transcript levels of *LcUbe2g1* in different tissues. RT-qPCR was performed to determine the relative expression of *LcUbe2g1* in different tissues of *Larimichthys crocea*, including brain, kidney, gills, head kidney, spleen, liver, fins and intestine. *β*-actin was used as an internal control to normalize the expression level; (**B**) Subcellular localization of *Lc*UBE2G1 protein in transfected *Larimichthys crocea* head kidney cells. The images were captured under confocal fluorescence microscopy. Scale bar = 50 µm.

**Figure 4 ijms-23-08298-f004:**
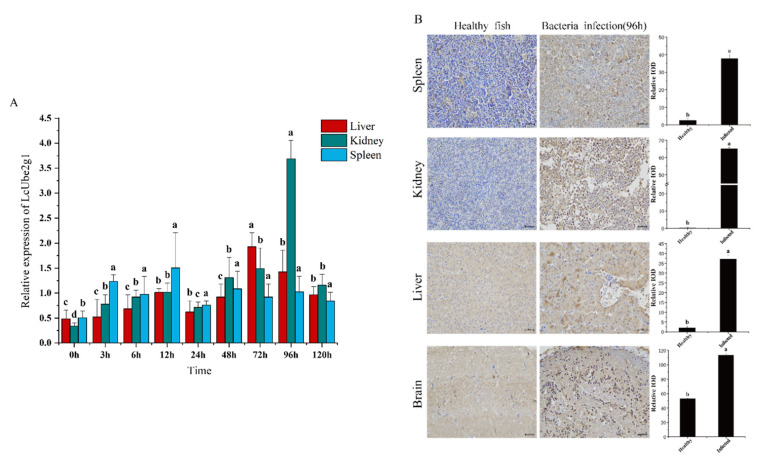
Defense response of UBE2G1 against *P.plecoglossicid*. (**A**) Time course expression of *LcUbe2g1* in different tissues after *P.plecoglossicid* infection. The relative expression levels of *LcUbe2g1* were determined by RT-qPCR in kidney, liver and spleen at 0 h, 3 h, 6 h, 12 h, 24 h, 48 h, 72 h, 96 h and 120 h after *P. plecoglossicid* challenge. *β-actin* was used as an internal control. Data are mean ± SE (*n* = 6 or 3). The letters a, b, c, d represent statistical significance (*p* < 0.05); (**B**) Immunohistochemical staining of UBE2G1 protein in spleen, kidney, liver and brain of large yellow croaker. Positive signal was indicated with brown stain.

**Figure 5 ijms-23-08298-f005:**
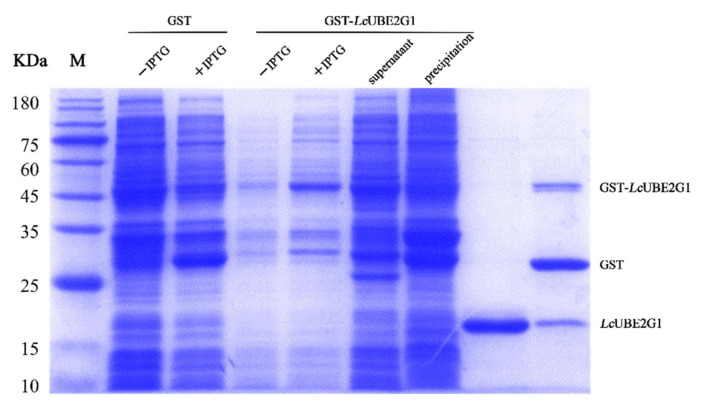
SDS-PAGE analysis of recombinant *Lc*UBE2G1 fusion protein.

**Figure 6 ijms-23-08298-f006:**
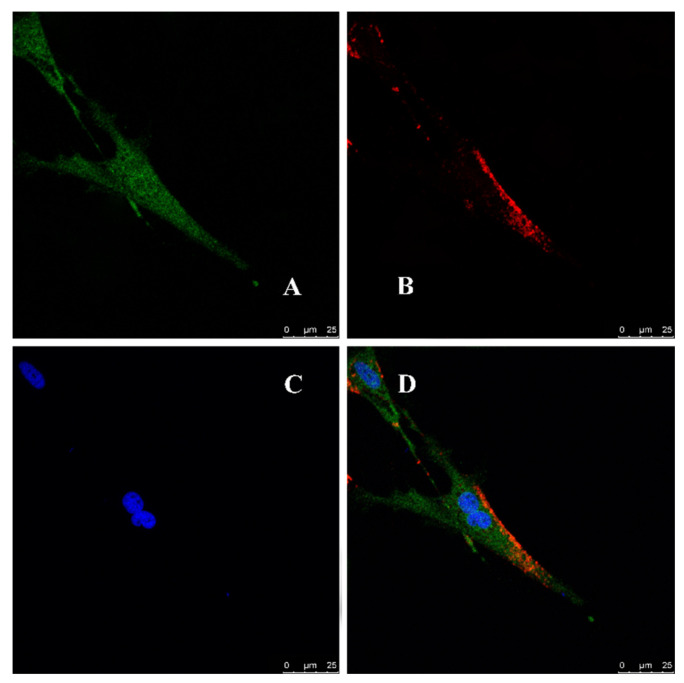
Co-localization of *Lc*UBE2G1 and ubiquitin in *Larimichthys crocea* head kidney cells detected by laser confocal microscopy. (**A**) *Lc*UBE2G1 displayed by Alexa Fluor 488-labeled secondary antibody (green); (**B**) Ubiquitin (Ub) displayed by Alexa Fluor 555-labeled secondary antibody (red); (**C**) Cell nuclei stained with DAPI (blue); (**D**) The combined graphs of (**A**–**C**).

**Figure 7 ijms-23-08298-f007:**
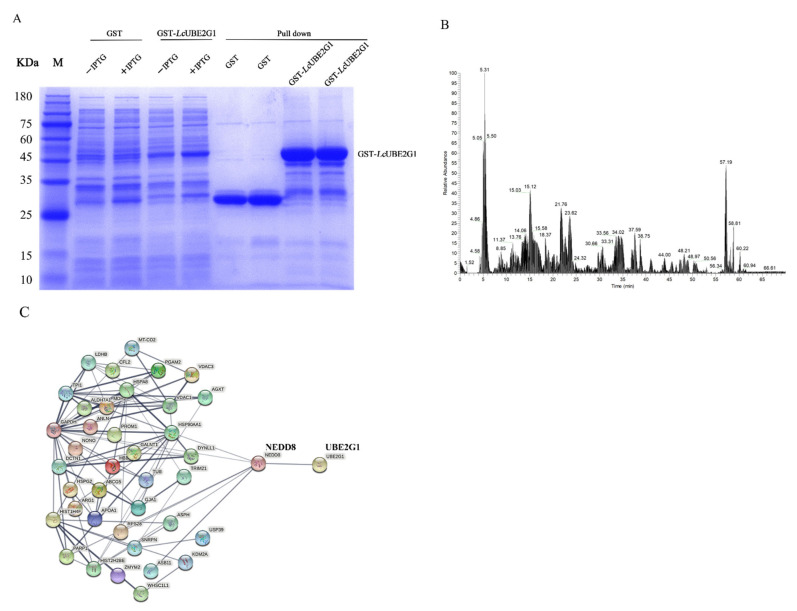
(**A**) SDS-PAGE analysis of GST pull-down results of recombinant *Lc*UBE2G1 fusion protein; (**B**) GST pull-down protein mass spectrometry results; (**C**) STRING analysis of proteins obtained by mass spectrometry.

**Table 1 ijms-23-08298-t001:** Identity analysis of amino acid sequences of *Ube2g1* in *Larimichthys crocea* and other species.

Species	Common Name	GenBank Accession Numbers	Identity (%)
*Larimichthys crocea*	Large yellow croaker	ON081957.1	100%
*Sparus aurata*	Sparus aurata	XP_030294309.1	99.41%
*Homo sapiens*	Human	AAH26288.2	97.06%
*Mus musculus*	Mouse	EDL12691.1	97.06%
*Gallus gallus*	Red Junglefowl	NP_001074352.1	97.06%
*Tetraodon nigroviridis*	Spotted green pufferfish	CAF98540.1	94.01%
*Takifugu rubripes*	Tiger Puffer	XP_003970748.1	93.98%
*Nibea albiflora*	Yellow drum	KAG8011671.1	86.62%
*Brachydanio rerio var*	Danio rerio	AAH71506.1	86.47%
*Xenopus tropicalis*	Xenopus tropicalis	NP_001017198.1	53.80%
*Melopsittacus undulatus*	Budgerigar	XP_005151071.1	53.17%

**Table 2 ijms-23-08298-t002:** Primers used in this experiment.

Primer Name	Sequences (5′–3′)	Annealing Temperature (°C)	Purpose
P1(*LcUbe2g1*-cF)	ATGACCGAGCCTCAGTCAGCGCTGT	62	cDNA cloning
P2(*LcUbe2g1*-cR)	TCACTCAAACGCAGTCTCTTGGCTT	58	
P3(*LcUbe2g1*-GST-F)	CCCCTGGGATCCCCGGAATTCATGACCGAGCCTCAGTCAGCG	75	Prokaryotic expression
P4(*LcUbe2g1*-GST-R)	GTCACGATGCGGCCGCTCGAGTCACTCAAACGCAGTCTCTTGG	74	
P5(*LcUbe2g1*-GFP-F)	CTACCGGACTCAGATCTCGAGATGACCGAGCCTCAGTCAGCG	73	Subcellular localization
P6(*LcUbe2g1*-GFP-R)	GTACCGTCGACTGCAGAATTCCCTCAAACGCAGTCTCTTGG	71	
*β*-actin-QF	TTATGAAGGCTATGCCCTGCC	54	qPCR-PCR
*β*-actin-QR	TGAAGGAGTAGCCACGCTCTGT	56	
*LcUbe2g1*-QF	CAAATGTGGACGCTGCAAAAG	53	qPCR-PCR
*LcUbe2g1*-QR	CCTGACGACAGGGTGGAAGA	56	

## Data Availability

Not applicable.

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
