# Peer review of "UBE2G1 Is a Critical Component of Immune Response to the Infection of Pseudomonas Plecoglossicida in Large Yellow Croaker (Larimichthys crocea)"

_ijms, 2022, doi:10.3390/ijms23158298_

Round 1

Reviewer 1 Report

The authors did a great job in reviewing the literature in the introduction section, without rambling. The methods are clearly presented and reproducible and they set their goals clearly. Results are very well presented and the figures are explanatory and of a good quality. The discussion is rather short but it is to the point. Some minor spelling changes in language have to be done throughout the manuscript

Author Response

Point1: Some minor spelling changes in language have to be done throughout the manuscript.

Response1: We thank the reviewer’s helpful suggestion. We carefully checked the manuscript for spelling errors and made  corrections.

Reviewer 2 Report

The paper describes the characterization of a gene LcUbe2g1 which is a predicted ubiquitylation enzyme responsible for innate cellular immunity against foreign proteins. Authors found this enzyme upregulated in a Pseudomonas plecoglossicida infection. This is an interesting study as ubiquitylation is usually associated with intracellular defense against viruses and not extracellular bacterial diseases. Rather than an innate immunity component, it is very much possible this upregulation is to protect important organs i.e., the brain and kidneys by recycling misfolded proteins during severe hemorrhagic ascites infection.

From a protein characterization point, this is a very systematic in silico study and has sufficient evidence for the functional category of the protein in question.

The colocalization studies confirm that this protein is present in both the cytoplasm and nucleus. The pull-down interaction with a negative regulator of ubiquitylation NEDD8 further confirms this protein’s role in DNA repair.

Minor comments

Human is misspelled as Homan in Table1

In figure2 phylogenetic tree has only one member in the bird and amphibian group therefore should be labeled as singular.

All in all, this is a good pioneering study on a ‘new’ protein. It would have been interesting if the authors discussed the relation of this protein to Pseudomonas plecoglossicida disease in more detail.

Decision: acceptable after extensively removing silly language errors. Possibly using MDPI proofreading service

Author Response

Point 1:Human is misspelled as Homan in Table1.

Response1: We thank the reviewer’s helpful suggestion. We have corrected the spelling error of human in Table 1.

Point 2: In figure2 phylogenetic tree has only one member in the bird and amphibian group therefore should be labeled as singular.

Response1: We thank the reviewer’s helpful suggestion.We have changed the two words in figure2 to the singular.